# Epicardial Adipose Tissue Thickness Is Related to Plaque Composition in Coronary Artery Disease

**DOI:** 10.3390/diagnostics12112836

**Published:** 2022-11-17

**Authors:** Soon Sang Park, Jisung Jung, Gary S. Mintz, Uram Jin, Jin-Sun Park, Bumhee Park, Han-Bit Shin, Kyoung-Woo Seo, Hyoung-Mo Yang, Hong-Seok Lim, Byoung-Joo Choi, Myeong-Ho Yoon, Joon-Han Shin, Seung-Jea Tahk, So-Yeon Choi

**Affiliations:** 1Department of Biochemistry and Molecular Biology, Ajou University School of Medicine, Suwon 16499, Republic of Korea; 2Department of Biomedical Sciences, Ajou University Graduate School of Medicine, Suwon 16499, Republic of Korea; 3Department of Cardiology, Ajou University Medical Center, Suwon 16499, Republic of Korea; 4Cardiovascular Research Foundation, New York, NY 10019, USA; 5Department of Biomedical Informatics, Ajou University School of Medicine, Suwon 16499, Republic of Korea; 6Office of Biostatistics, Medical Research Collaborating Center, Ajou Research Institute for Innovative Medicine, Ajou University Medical Center, Suwon 16499, Republic of Korea

**Keywords:** lipid core plaque, lipid core burden index, epicardial adipose tissue, near-infrared spectroscopy

## Abstract

(1) Background: Currently, limited data are available regarding the relationship between epicardial fat and plaque composition. The aim of this study was to assess the relationship between visceral fat surrounding the heart and the lipid core burden in patients with coronary artery diseases; (2) Methods: Overall, 331 patients undergoing coronary angiography with combined near-infrared spectroscopy and intravascular ultrasound imaging were evaluated for epicardial adipose tissue (EAT) thickness using transthoracic echocardiography. Patients were divided into thick EAT and thin EAT groups according to the median value; (3) Results: There was a positive correlation between EAT thickness and *max*LCBI_4mm_, and *max*LCBI_4mm_ was significantly higher in the thick EAT group compared to the thin EAT group (437 vs. 293, *p* < 0.001). EAT thickness was an independent predictor of *max*LCBI_4mm_ ≥ 400 along with age, low-density lipoprotein-cholesterol level, acute coronary syndrome presentation, and plaque burden in a multiple linear regression model. Receiver operating characteristic curve analysis showed that EAT thickness was a predictor for *max*LCBI_4mm_ ≥ 400; (4) Conclusions: In the present study, EAT thickness is related to the lipid core burden assessed by NIRS-IVUS in patients with CAD which suggests that EAT may affect the stability of the plaques in coronary arteries.

## 1. Introduction

Visceral adiposity promotes both local and systemic inflammation, and its presence influences the development of metabolic as well as cardiovascular diseases [1]. Epicardial adipose tissue (EAT) is fat deposition around the heart; it mediates cardiac function and influences development of coronary artery atherosclerosis [2,3]. Although several studies have demonstrated the association between EAT thickness and the severity of coronary artery disease (CAD) [4,5,6,7], limited data are available for the relationship between EAT and plaque composition.

Intravascular imaging modalities are considered to be the best tools to assess the vulnerability of coronary plaques in patients with CAD. Catheter-based near-infrared spectroscopy intravascular ultrasound (NIRS-IVUS) identifies the extent of coronary artery lipid-rich plaque as measured by the lipid core burden index (LCBI) [8,9,10]. A high LCBI identifies a vulnerable plaque [11,12] and is associated with future cardiovascular events [13,14,15,16,17,18]. Maddler and colleagues demonstrated that a plaque with a high LCBI detected by NIRS was responsible for acute coronary syndrome [11]. Waksman and colleagues showed the ability of NIRS to detect plaque vulnerability related with future cardiac events on the patient and on non-culprit plaque levels with a prespecified cutoff of the LCBI [18]. Thus, understanding the relationship between epicardial fat and the plaque component may reveal the role of epicardial fat on the plaque vulnerability and its value as a predictor of cardiovascular events. The aim of the present study is to assess the relationship between the echocardiographic EAT thickness and the lipid core burden index in patients with CAD.

## 2. Patients and Methods

### 2.1. Study Population

Patients imaged with NIRS-IVUS at Ajou University Hospital (Suwon, Republic of Korea) were enrolled in a dedicated registry (the Ajou NIRS-IVUS registry). A flowchart of this study is shown in Figure 1. Among 433 patients imaged with NIRS-IVUS from February 2015 to January 2020, 331 patients were enrolled in this study. Exclusion criteria for the 102 patients were in-stent restenosis (*n* = 54), poor image quality (*n* = 7), no echocardiography within a week before or after NIRS-IVUS evaluation (*n* = 12), and patients who had a repeated examination during follow-up (*n* = 29). The data of NIRS-IVUS, echocardiography, and medical records were retrospectively reviewed in an independent manner (NIRS-IVUS: J.J, B.-J.C, and U.-R. J; echocardiogram: J.-S.P and J.-H.S; medical records: B.-J.C and S.-Y.C). This study was approved by the Institutional Review Board of Ajou University Hospital (AJOUIRB-MDB-2019-094, Suwon, Republic of Korea).

### 2.2. Echocardiography 

EAT on two-dimensional transthoracic echocardiography appears as a hypoechoic space between the linear hyperechoic parietal pericardium and the right ventricle epicardium [4]. Recordings of three cycles in the standard parasternal long axis view at the basal left ventricular level were obtained. Maximum EAT thickness was measured perpendicular to the free wall of the right ventricle on the parasternal long axis at the end of diastole (Figure 2A). The interclass correlation coefficients were 0.983 (95% confidence interval (CI) 0.964-0.992, *p* < 0.0001) and 0.995 (95% CI 0.990-0.998, *p* < 0.0001) for the inter-and intra-observer variability of the EAT thickness measurements, respectively, indicating good reproducibility and feasibility. Patients were divided into two groups; thin EAT group and thick EAT group according to the median value of EAT thickness. Clinical data, coronary angiography, and NIRS-IVUS in the thick versus thin EAT groups were compared between the two groups.

### 2.3. Coronary Angiography and NIRS-IVUS

Coronary angiography was obtained using standard techniques, and all angiograms were analyzed in a blinded fashion at the Ajou University imaging analysis core laboratory using Cardiovascular Angiography Analysis System II (CAAS II, Pie Medical, Maastricht, The Netherlands) (Figure 2B). After diagnostic coronary angiography, a symptom-related or suspected symptom-related coronary artery was identified as the target vessel for NIRS-IVUS imaging; and a second and third vessel was imaged as appropriate. The NIRS-IVUS catheter (TVC Imaging System, InfraRedx, Burlington, MA, USA) was advanced into each vessel; and simultaneous IVUS images and co-registered NIRS measurements were acquired during automatic single rotational pullback at 0.5 mm per second and interpreted as previously described [8,9,19,20]. Quantitative IVUS analysis was performed along the entire length of the vessel on cross-sectional images spaced 1mm apart. External elastic membrane cross-sectional area, lumen cross-sectional area, and minimal lumen area were measured at the smallest lumen cross-sectional area site. Plaque burden was defined as (plaque cross-sectional area [external elastic membrane cross-sectional area—lumen cross-sectional area] divided by external elastic membrane cross-section area) × 100 (Figure 2C,D). Lesion length was calculated from IVUS frame counts using automatic transducer pullback. The chemogram containing the NIRS data was displayed in a digital color-coded map of intensity and location of lipid from the luminal surface with the x-axis representing pullback distance within the scanned artery and the y-axis rotation from 0 to 360 degrees on the two-dimensional rectangular map. For each pixel, red color is no probability of lipid; and yellow color is a high probability of lipid. Lipid content is described as LCBI, the percentage of yellow pixels in a region-of-interest multiplied by 1000 with a range from 0 to 1000. The largest maximum lipid core burden index in any 4mm segment (*max*LCBI_4mm_) was used as a surrogate for plaque vulnerability (Figure 2E). In case of multiple lesions or multivessel disease, one culprit lesion per patient was selected by a physician’s clinical judgement based on multivessel intracoronary imaging mean luminal area, plaque burden, and LCBI values used for all analysis were obtained from a single culprit lesion. If a patient had multiple potential culprits, the one with the largest LCBI_4mm_ was used.

### 2.4. Statistical Analysis 

All statistical analyses were performed using SPSS 23.0 (SPSS in., Chicago, IL, USA). Data are expressed as median value (interquartile range (IQR)) for continuous variables and as numbers and percentages for categorical variables. Comparisons of continuous and categorical variables were performed using the Wilcoxon Rank-sum test/ANOVA and chi-square test, respectively. Univariate and multiple logistic regression analysis was performed to assess independent factors such age, smoking, body mass index, diabetes mellitus, hypertension, high density lipoprotein (HDL) cholesterol, low density lipoprotein (LDL) cholesterol, acute coronary syndrome (ACS) presentation, mean luminal area, plaque burden, and EAT thickness that were potentially related to *max*LCBI_4mm_. A receiver operation characteristic (ROC) curve analysis was used to assess the best cut-off value of EAT thickness to predict *max*LCBI_4mm_ ≥ 400 with maximal accuracy. A *p*-value of less than 0.05 was considered statistically significant.

## 3. Results

### 3.1. Baseline Characteristics

Baseline patient characteristics have been summarized in Table 1. The median age of patients was 58 years (IQR 51–64), and 76.7% (254 patients) were males. Overall, 63.7% of patients present with ACS. Among the 331 patients, only one vessel was scanned using NIRS-IVUS in 235 patients (71%), two vessels in 44 patients (13%), and three vessels in 52 patients (15.7%). Symptom-related vessels were 284 (85.8%) left anterior descending arteries, 15 (4.5%) left circumflex arteries, and 32 (9.7%) right coronary arteries. The median maximum EAT thickness was 2.24 (IQR 1.03–3.36) mm. Eighty-four (25.4%) patients had dyslipidemia, and all patients diagnosed with dyslipidemia were taking statins, and the use of a combination therapy of statin and ezetimibe was 3.8% of the total patients. Patients were divided into two groups: thin EAT group (<2.24 mm, *n* = 164) and thick EAT group (≥2.24 mm, *n* = 167). Patients in the thick EAT group were significantly older and more often males and had a higher BMI and lower HDL cholesterol than patients in the thin EAT group. Additionally, the thick EAT group were more often diabetic and hypertensive and more often presented with ACS. 

### 3.2. Angiographic and NIRS-IVUS Characteristics

Angiographic and NIRS-IVUS findings have been summarized in Table 2. Triple vessel disease was slightly more common in the thick EAT group compared with the thin EAT group. Compared with the thin EAT group, the thick EAT group had a significantly worse diameter stenosis on angiography and a larger plaque burden, smaller mean luminal area, and higher *max*LCBI_4mm_ value on NIRS-IVUS imaging.

### 3.3. EAT Thickness Has a Positive Correlation with maxLCBI_4mm_

The median *max*LCBI_4mm_ was 386 (IQR 244–528). As shown in Figure 3, EAT thickness had a positive correlation with *max*LCBI_4mm_ (*r* = 0.420, *p* < 0.001). The *max*LCBI_4mm_ increased according to the interquartile range of EAT (Appendix A).

### 3.4. EAT Thickness Predicts maxLCBI_4mm_ ≥ 400

To evaluate the effect of variables on plaque composition, logistic regression analysis was performed. Comorbidities, such as DM, hypertension, and dyslipidemia, and chronic renal disease (CKD) which can affect the plaque composition were included in this analysis. In a univariate logistic regression analysis model, age (*p* < 0.001), dyslipidemia (*p =* 0.046), LDL cholesterol (*p* = 0.008), ACS presentation (*p* < 0.001), mean luminal area (*p* < 0.001), plaque burden (*p* < 0.001), and EAT thickness (*p* < 0.001) were shown to be predictors of *max*LCBI_4mm_ ≥ 400 (Table 3). However, DM, hypertension, and CKD were not related with *max*LCBI_4mm_ ≥ 400. Additionally, in a multiple logistic regression analysis model, age (*p* = 0.017), LDL cholesterol (*p* = 0.003), ACS presentation (*p* < 0.001), plaque burden (*p* = 0.048), and EAT thickness (*p* < 0.001) were independent factors for *max*LCBI_4mm_ ≥ 400 (Table 3). ROC curve was used to determine the EAT thickness that predicted *max*LCBI_4mm_ ≥ 400. The best cut-off value was 2.245 [area under the curve (AUC) = 0.677, sensitivity of 65.4 %, and specificity of 66.0 %] (Figure 4). 

The best cut-off value of EAT thickness for *max*LCBI_4mm_ was 2.245 (AUC = 0.677, 95% CI: 0.618–0.735, sensitivity: 65.4%, specificity: 66.0%). AUC, area under curve; CI, confidence interval; EAT, epicardial adipose tissue; *max*LCBI_4mm_, maximum lipid core burden index in any 4mm segment.

## 4. Discussion

### 4.1. Local Fat and Cardiovascular Diseases 

There is growing evidence that excess body fat accumulation is associated with metabolic and cardiovascular risks [21,22]. Ectopic fat depots, defined by fat stored in locations not classically related with adipose storage, are thought to be a main cause of various cardiovascular diseases by acting as an active organ secreting various inflammatory mediators [1,23]. Recently, the roles of local ectopic fat on epicardial or surrounding coronary vessels are getting more attention because most of them are located adjacent to the heart and vessels and may directly affect these organs resulting in coronary atherosclerosis, arrhythmia, and heart failure [24,25,26]. One possible explanation for this is its paracrine effect of local ectopic fat; it is metabolically active and produces a variety of bioactive molecules that can have a significant impact directly to the heart or vessels [27,28]. Especially, EAT, the most abundant fat among local fats and located near the heart, is thought to be related to not only atherogenesis, but also amplification of vascular inflammation, plaque instability by apoptosis, and neovascularization [29,30]. We previously reported that echocardiographic EAT thickness is associated with the extent of coronary atherosclerosis and unstable clinical presentation [7]. Our current data also show that the thickness of local ectopic fat, EAT, is related to *max*LCBI_4mm_ of the culprit lesion. On the other hand, factors including BMI and diabetes mellitus which may represent systemic ectopic fat did not show a correlation with *max*LCBI_4mm_ of the culprit lesion in univariate logistic regression. This suggests that the local effect of fat tissue might be more important than the systemic effect in the pathogenesis of coronary artery disease and plaque vulnerability. Attesting to this, there was a recent large-scale study showing that perivascular fat adjacent to the coronary arteries predicts cardiovascular risk using computed tomography angiography (the CRISP CT study) [31]. 

### 4.2. Assessment of Atherosclerotic Plaque

Numerous biomarkers associated with lipid metabolism and inflammation have been identified as novel targets to monitor atherosclerosis and cardiovascular risk [32]. Conventional markers, such as troponins and creatine kinases, are specific molecular features of atherosclerosis pathogenesis [33,34]. Well-known inflammatory markers, *hs*-CRP and IL-6 were also upregulated during atherosclerosis progression [35]. Moreover, soluble ST2 (sST2), a soluble receptor for IL-33, is an emerging marker to indicate the atherosclerotic burden. sST2 can bind with IL-33 and hinders cardioprotective effects of IL-33 [36,37]. Scicchitano and colleagues revealed that sST2 can also act as a prognostic factor of all-cause mortality after carotid plaque endarterectomy [38]. Demyanet and colleagues also showed that sST2 is related to plaque vulnerability and rupture in coronary acute coronary syndrome [39]. The most frequently used imaging methodologies to assess plaque characteristics are coronary computer tomography, angioscopy, IVUS, radiofrequency-IVUS [virtual histology (VH)-IVUS], optical coherence tomography (OCT), and NIRS [40]. These modalities can identify culprit lesions at high risk of peri-procedural myocardial injury due to distal embolization [41]; non-culprit lesions at high risk of future cardiovascular events [17,18]. Compared to coronary computer tomography, intravascular imaging has been considered the best tool to detect vulnerable plaque features based on its higher resolution and diagnostic accuracy [42]. Previously it has been reported that VH-thin cap fibroatheroma (TCFA) (by VH-IVUS) or OCT-TCFA/lipid-rich plaque by OCT was more commonly found in patients with ACS than in those with stable angina and was related with procedure-related myocardial injury in culprit lesion [43,44]. Furthermore, VH-TCFA in the PROSPECT, a prospective natural observation study by using VH-IVUS [45] and plaques with OCT-derived high-risk features in the CLIMA study using OCT were associated with future cardiovascular events [46]. High risk features in PROSPECT were a VH-TCFA, mean luminal area < 4 mm [2], and plaque burden > 70% [45]. High risk features in CLIMA were a small mean luminal area, thin fibrous cap morphology, lipid arc circumferential extension, presence of OCT-defined macrophages [46]. OCT-derived LCBI_4mm_ ≥ 400 was another high-risk feature in a secondary analysis of the CLIMA study [47]. In this study, we conducted a NIRS-IVUS imaging evaluation to assess plaque vulnerability. The combined NIRS-IVUS imaging presents a color map of the lumen of the target vessel and shows the location and amount of lipid content within the plaque [48]. NIRS-IVUS accurately detects a lipid rich plaque in histologically validated plaques, but the greater advantage is that it provides information on the distribution and amount of lipid content in the vessel or the region-of-interest and is measured automatically compared to the obligatory operator interpretation of VH-IVUS or OCT [8,43,44,49,50,51]. Previous studies demonstrated that NIRS-IVUS-yellow plaque was frequently detected in patients with ACS compared to patients with stable angina and NIRS-detected high lipid content was related to periprocedural complication [20,33]. Recently two prospective observational studies using NIRS-IVUS, the LRP study and the PROSPECT II study, demonstrated that NIRS-yellow plaque was associated with future cardiovascular events [18,52]. In the LRP study Waksman and colleagues demonstrated that *max*LCBI_4mm_ more than 400 was an independent predictor for non-culprit MACE in both patient and lesion levels^18^. In the PROSPECT II study, Erlinge and colleagues showed that non-obstructive lesions with a high lipid content (*max*LCBI_4mm_ ≥ 324.7 [median value]) and large plaque burden (≥70%) were at increased risk for future adverse cardiac outcomes [52]. Our data also showed that EAT thickness was related to plaque burden; patients in the thick EAT group had a larger plaque burden than patients in the thin EAT group (*p* = 0.013). Additionally, plaque burden was an independent predictor of *max*LCBI_4mm_ ≥ 400 both in univariate and multivariate analyses. To our current knowledge, this is the first study which showed a correlation between the NIRS-IVUS findings and echocardiographic EAT thickness. It is consistent with our previous data that EAT thickness measured using VH-IVUS correlated with plaque vulnerability [53]. Therefore, echocardiographic EAT thickness may predict the stability of the plaques in coronary artery diseases.

### 4.3. Limitations

The present study has several limitations. First, this was a retrospective observational study conducted in a single Asian center and included a relatively small number of patients. Since this report depends on a review of charts, this association should be verified from more patients in prospective studies to exclude the effects of confounding variables (previous pharmacological treatments, comorbidities, and so on) and clarify their relationship with others affecting the plaque. Furthermore, to evaluate the prognostic role of EAT, long-term prospective studies are needed for the risk level prediction of future cardiovascular events. Second, EAT thickness measured by transthoracic echocardiography may not be the best method to quantify the amount of EAT, although previous studies showed positive correlation among EAT values obtained from thoracic echocardiography, computed tomography, or magnetic resonance imaging [54], and echocardiographic EAT thickness is a simple and practical method in clinical practice [3,55,56]. Third, NIRS-IVUS was not used to image all coronary arteries unlike the three vessel PROSPECT and PROSPECT II studies [45,52]. 

## 5. Conclusions

This study demonstrated that echocardiographic EAT thickness was highly correlated with a *max*LCBI_4mm_ value assessed by NIRS-IVUS imaging. Moreover, EAT thickness was an independent predictor of *max*LCBI_4mm_ ≥ 400 in patients with CAD. Our results suggest that EAT might have a specific role both in future cardiac risk prediction and plaque vulnerability assessment. 

## Figures and Tables

**Figure 1 diagnostics-12-02836-f001:**
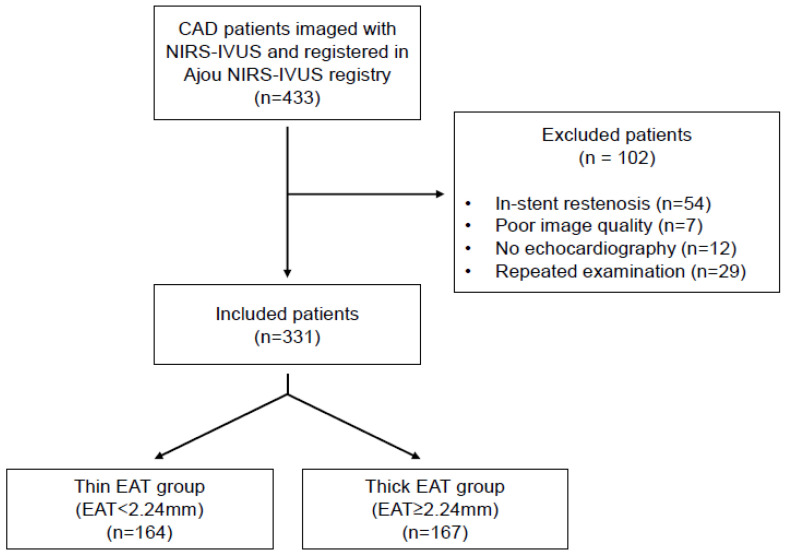
A flowchart of this study.

**Figure 2 diagnostics-12-02836-f002:**
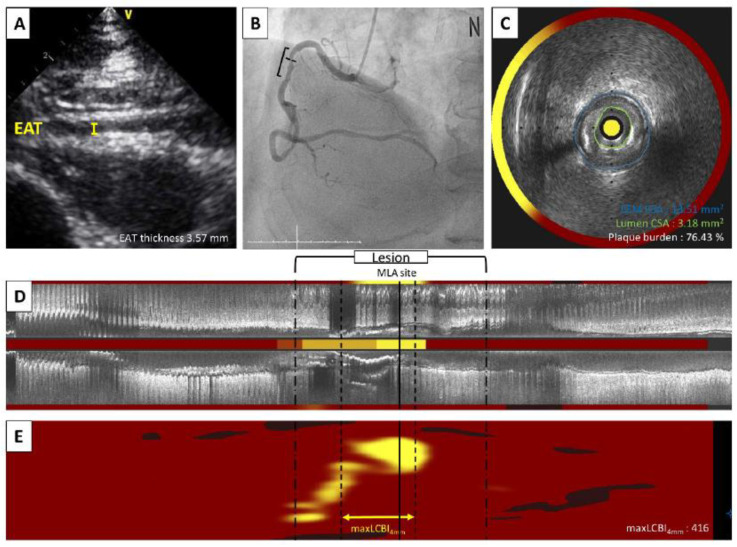
A representative case of a 68-year-old patient with unstable angina was evaluated by echocardiogram and NIRS-IVUS imaging. (**A**)Transthoracic echocardiography revealed EAT was 3.57 mm. (**B**) Right coronary artery has an intermediate stenosis at proximal part. (**C**) Cross-sectional NIRS-IVUS image showed 3.18 mm^2^ in lumen area, 76.4% of plaque burden and a quarter of yellow color in circumferential chemogram at the MLA site. (**D**) Longitudinal image with block chemogram and (**E**) color-coded map revealed a lipid rich plaque (yellow) with 416 *max*LCBI_4mm_. CSA, cross sectional area; EAT, epicardial adipose thickness; EEM, external elastic membrane; IVUS, intravascular ultrasound; *max*LCBI_4mm_, maximum lipid core burden index in any 4 mm segment; MLA, minimum lumen area, NIRS, near-infrared spectroscopy.

**Figure 3 diagnostics-12-02836-f003:**
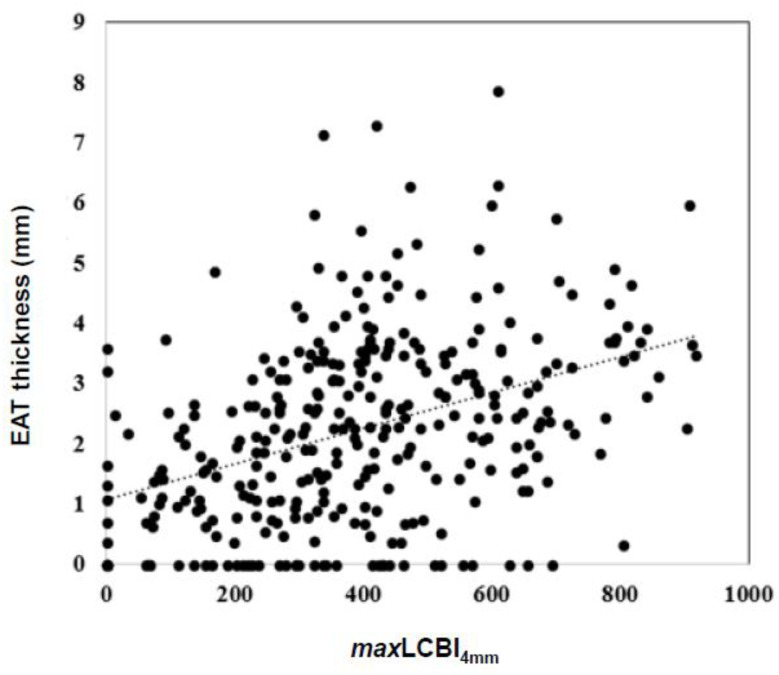
The relationship between EAT and *max*LCBI_4mm_. There was a positive correlation between *max*LCBI_4mm_ and EAT (*r* = 0.420, *r^2^* = 0.176, and *p* < 0.001). *p* value and coefficients are obtained from simple linear regression analysis. EAT, epicardial adipose tissue; *max*LCBI_4mm_, maximum lipid core burden index in any 4 mm segment.

**Figure 4 diagnostics-12-02836-f004:**
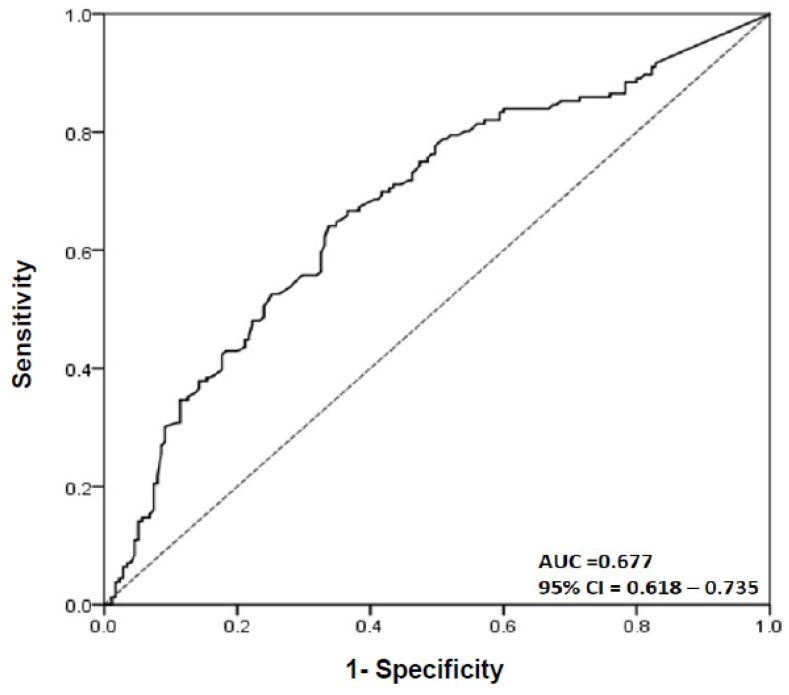
The ROC curve for EAT thickness to predict *max*LCBI_4mm_ ≥ 400.

**Table 1 diagnostics-12-02836-t001:** Patient characteristics of this study.

	Total (*n* = 331)	Thin EAT (*n* = 164)	Thick EAT (*n* = 167)	*p* Value
Age, year (IQR)	58 (51–64)	55.5 (48–61)	60 (55–68)	<0.001 ^1^
Male, *n* (%)	254 (76.7)	39 (84.8)	115 (68.9)	<0.001 ^2^
Smoker, *n* (%)	106 (32.0)	56(34.1)	50 (29.9)	0.181 ^2^
BMI, kg/m^2^, (IQR)	25.01 (23.31–27.35)	24.51 (22.95–26.45)	25.712 (23.94–27.86)	<0.001 ^1^
DM, *n* (%)	91 (27.5)	36 (22.0)	55 (32.9)	0.025 ^2^
Hypertension, *n* (%)	168 (51.0)	68 (41.5)	100 (60.2)	0.001 ^2^
CKD *, *n* (%)	32 (9.7)	4 (2.4)	28 (1.2)	0.001 ^2^
Previous PCI, *n* (%)	31 (9.4)	14 (8.5)	17 (10.2)	0.608 ^2^
Stroke, *n* (%)	9 (2.7)	3 (1.8)	6 (3.6)	0.502 ^2^
PAOD, *n* (%)	2 (0.6)	0 (0.0)	2 (1.2)	0.499 ^2^
Dyslipidemia, *n* (%)	84 (25.4)	37 (22.6)	47 (28.1)	0.298 ^2^
Total cholesterol, mg/dL (IQR)	173 (143–204)	176 (144–206)	172 (143–199)	0.747 ^1^
Triglyceride, mg/dL (IQR)	118 (84–178)	114 (79–156)	125 (87–192)	0.120 ^1^
HDL cholesterol, mg/dL (IQR)	44 (38–53)	46 (40–55)	43 (36–50)	0.001 ^1^
LDL cholesterol, mg/dL (IQR)	102 (72–131)	104 (76–133)	102 (71–127)	0.988 ^1^
LVEF, % (IQR)	65 (58–71)	65 (59–72)	64 (58–70)	0.265 ^1^
ACS, *n* (%)	211 (63.7)	95 (57.9)	121 (69.5)	0.019 ^2^

* Creatinine clearance <60mL/min as calculated at baseline by the Cockcroft-Gault equation. ^1^
*p* values were obtained from Wilcoxon Rank-sum test. ^2^
*p* values were obtained from the chi-square test. Values are expressed as the median (IQR) or the percentage (%). ACS, acute coronary syndrome; CKD, chronic kidney disease; DM, diabetes mellitus; EAT; epicardial adipose tissue; HDL, high density lipoprotein; HTN, hypertension; IQR, interquartile range; LDL, low density lipoprotein; LVEF, left ventricular ejection fraction; *max*LCBI_4mm_, maximum lipid core burden index in any 4 mm segment; PAOD, peripheral artery obstructive disease; PCI, percutaneous coronary intervention.

**Table 2 diagnostics-12-02836-t002:** Angiographic and NIRS-IVUS characteristics.

	Total (*n* = 331)	Thin EAT (*n* = 164)	Thick EAT (*n* = 167)	*p* Value
Symptom-related vessel, *n* (%)				0.052 ^2^
LAD	284 (85.8)	143 (87.2)	141 (84.4)	-
LCX	15 (4.5)	3 (1.8)	12 (7.2)	-
RCA	32 (9.7)	18 (11.0)	14 (8.4)	-
Vessel disease, *n* (%)				0.036 ^2^
Intermediate	34 (10.3)	24 (14.6)	10 (6.0)	-
1VD	132 (39.9)	63 (38.4)	69 (41.3)	-
2VD	98 (29.6)	50 (30.5)	48 (28.7)	-
3VD	67 (20.2)	27 (16.5)	40 (24.0)	-
Angiographic diameter stenosis, % (IQR)	85.0 (74–95)	84.0 (70.0–92.0)	88.0 (76.0–98.0)	0.007 ^1^
NIRS-IVUS analysis				
Minimum lumen area, mm^2^ (IQR)	2.80 (2.14–3.67)	2.94 (2.30–4.20)	2.50 (2.07–3.50)	<0.001 ^1^
PB, % (IQR)	75.3 (64.2–83.2)	72.8 (59.8–82.2)	76.5 (69.1–84.2)	0.013 ^1^
*max*LCBI_4mm_ (IQR)	386 (244–528)	293 (154–429)	437 (335–606)	<0.001 ^1^
*max*LCBI_4mm_ ≥ 400, *n* (%)	156 (47.1)	53 (32.3)	103 (61.7)	<0.001 ^2^

Values are expressed as the median (IQR) or the percentage (%). ^1^
*p* values were obtained from Wilcoxon Rank-sum test. ^2^
*p* values were obtained from the chi-square test. EAT, epicardial adipose tissue; IVUS, intravascular ultrasound; IQR, interquartile range; LAD, left anterior descending artery; LCX, left circumflex artery; maxLCBI_4mm_, maximum lipid core burden index any in 4 mm segment; NIRS, near-infrared spectroscopy; PB, plaque burden; RCA, right coronary artery; VD, vessel disease.

**Table 3 diagnostics-12-02836-t003:** Univariate and multivariate logistic regression analysis for independent predictors of *max*LCBI_4mm_ ≥ 400.

Variables	Univariate Logistic Regression Analysis					Multiple Logistic Regression Analysis				
	B ± SE	Odds Ratio	95% CI		*p* Value	B ± SE	Odds Ratio	95% CI		*p* Value
			Lower	Upper				Lower	Upper	
Age	0.040 ± 0.011	1.041	1.018	1.065	<0.001	0.031 ± 0.013	1.032	1.006	1.059	0.017
Male	0.312 ± 0.221	1.377	0.826	2.303	0.221	-	-	-	-	-
Smoking	0.206 ± 0.221	1.228	0.798	1.895	0.351	-	-	-	-	-
BMI	0.022 ± 0.036	1.022	0.952	1.098	0.543	-	-	-	-	-
DM	0.189 ± 0.247	1.208	0.745	1.961	0.443	-	-	-	-	-
HTN	0.248 ± 0.221	1.282	0.831	1.980	0.262	-	-	-	-	-
CKD *	0.413 ± 0.375	1.511	0.725	3.150	0.271					
Dyslipidemia	0.505 ± 0.254	1.658	1.008	2.726	0.046	0.373 ± 0.292	1.452	0.819	2.574	0.202
HDL	−0.013 ± 0.009	0.987	0.969	1.005	0.149	-	-	-	-	-
LDL	0.008 ± 0.003	1.008	1.002	1.013	0.008	0.010 ± 0.003	1.010	1.003	1.017	0.003
ACS	1.065 ± 0.242	2.902	1.818	4.695	<0.001	1.003 ± 0.282	2.728	1.568	4.745	<0.001
MLA	−0.271 ± 0.075	0.763	0.652	0.876	<0.001	−0.028 ± 0.088	0.972	0.818	1.156	0.752
PB	0.034 ± 0.007	1.034	1.020	1.050	<0.001	0.021 ± 0.012	1.018	1.001	1.037	0.048
EAT	0.366 ± 0.078	1.442	1.244	1.686	<0.001	0.340 ± 0.090	1.405	1.177	1.678	<0.001

* Creatinine clearance < 60 mL/min as calculated at baseline by the Cockcroft-Gault equation. ACS, Acute coronary syndrome; BMI, body mass index; CKD, chronic kidney disease; DM, diabetes mellitus; EAT, epicardial adipose tissue thickness; HDL, high density lipoprotein cholesterol; HTN, hypertension; LDL, low density lipoprotein cholesterol; *max*LCBI_4mm_, maximum lipid core burden index in any 4mm segment; MLA, minimum lumen area; PB, plaque burden.

## Data Availability

Not applicable.

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
