# Peer review of "Epicardial Adipose Tissue Thickness Is Related to Plaque Composition in Coronary Artery Disease"

_diagnostics, 2022, doi:10.3390/diagnostics12112836_

Round 1

Reviewer 1 Report

To:

Editorial Board

Diagnostics

Title: “Epicardial Adipose Tissue Thickness is Related to Plaque Composition in Coronary Artery Disease”

Dear Editor,

I read this paper and I think that:

-       Was this a retrospective study? The retrospective nature of this paper is a limitation of this paper. Please discuss such a point in a dedicated limitation section.

-       A flow chart of the study should be added in order to improve the readability of the text.

-       The reproducibility of the techniques should be provided. Please calculate inter and intraobserver variability coefficients. Please provide.

-       All pharmacological treatments should be declared and computed in the final regression model. Please provide.

-       There is no mention about comorbidities of the patients. This is fundamental information as it can impact on results. Please report data.

-       The role of ST2 in the context of the atherosclerotic plaques should be stressed. This should be deeply discussed. Authors can consider and discuss the paper from Scicchitano P et al. J Clin Med. 2022 May 31;11(11):3142.

Reviewer 2 Report

Overall, a good study showing the significant correlation between EAT thickness and lipid core burden. Some specific comments are listed for improvement:

1. Introduction: 2nd para, authors cited: high LCBI identifies a vulnerable plaque [11,12] and is associated with future cardiovascular events [13-47 18]. Authors are suggested to expand the literature review a little more with key findings, and also highlight the research gap in the current literature. The current version did not provide readers with information about why this study was conducted or what it contributed. 

2. Methods: section 2.2, who performed EAT measurements? how to resolve inter- or intra-observer variability?

3. Results: Tables 1-3. only keep the heading in the tables while moving these explanation to the foot note. 

4. Discussion: Good discussion with correlation of your study findings to the literature. Computer tomography-change to computed tomography throughout the manuscript. 

Author Response

Response to Reviewer #2: Manuscript ID: diagnostics-2011787

Overall, a good study showing the significant correlation between EAT thickness and lipid core burden. Some specific comments are listed for improvement:

We really appreciate the Reviewer’s careful review and valuable comments to our manuscript. We have attempted to incorporate the Reviewer’s suggestions in the current revision, and we strongly believe that these suggestions have improved the overall scientific content of this work. Changes from the previous manuscript was highlighted in the revised version. Responses to the scientific and statistical recommendations are given below:

R1. Introduction: 2nd para, authors cited: high LCBI identifies a vulnerable plaque [11,12] and is associated with future cardiovascular events [13-18]. Authors are suggested to expand the literature review a little more with key findings, and also highlight the research gap in the current literature. The current version did not provide readers with information about why this study was conducted or what it contributed. 

A1. As the reviewer recommended, we have expanded recent findings and clarified the research gap and purpose of this study in 1. Introduction as follow: “Maddler and colleagues demonstrated that a plaque with high LCBI detected by NIRS was responsible for acute coronary syndrome [11]. Waksman and colleagues showed the ability of NIRS to detect plaque vulnerability related with future cardiac events on the patient and on non-culprit plaque levels with a prespecified cutoff of the LCBI [18]. Thus, knowing the relationship between epicardial fat around the heart and plaque component may help to understand the role of epicardial fat on the plaque vulnerability and its value as a predictor of cardiovascular events.”

R2. Methods: section 2.2, who performed EAT measurements? how to resolve inter- or intra-observer variability?

A2. Two experienced physicians (J.-S.P and J.-H.S) independently measured maximum EAT thickness on two-dimensional transthoracic echocardiography. We calculated inter and intra-observer variability from 30 random samples. The absolute values of the mean paired differences were 0.049±0.350 (p=0.447) and 0.044 ±0.184 (p=0.193) for inter- and intra-observer measurements of EAT. Interclass correlation coefficients were 0.983 [95% confidence interval (CI) 0.964-0.992, p<0.0001) and 0.995 (95% CI 0.990-0.998, p<0.0001) for inter and intra-observer variability indicating good reproducibility and feasibility. Our results are consistent with previous studies [Parisi et al., Nutr Metab Cardiovasc Dis 2020;30:99]. We have provided the value of interclass correlation coefficients for EAT measurements within 2.2. Echocardiography in Methods as follow: Interclass correlation coefficients were and 0.983 [95% confidence interval (CI) 0.964-0.992, p<0.0001] and 0.995 (95% CI 0.990-0.998, p<0.0001) for inter-and intra-observer variability of the EAT thickness measurements, indicating good reproducibility and feasibility.”

R3. Results: Tables 1-3. only keep the heading in the tables while moving these explanation to the foot note. 

A3. As the reviewer recommended, we have moved explanations of the table to the foot note.

R4. Discussion: Good discussion with correlation of your study findings to the literature. Computer tomography-change to computed tomography throughout the manuscript. 

A4. We deeply appreciate the reviewer’s careful comment about spelling mistake. We have changed “computer tomography” to “computed tomography” throughout the manuscript.  

Round 2

Reviewer 1 Report

Gli autori hanno risposto bene ai miei commenti precedenti. la carta è migliorata molto